# Early Response of 'Mexican' Lime, 'Fina' Clementine Mandarin, and 'Campbell' Valencia Orange on Selected Rootstocks Grown under Fertigation Practices in an Oxisol in Puerto Rico

**Rebecca Tirado-Corbalá** \*, **Elvin Román-Paoli, Alejandro E. Segarra-Carmona, Consuelo Estévez de Jensen and Dania Rivera-Ocasio**

Agro-Environmental Sciences Department, University of Puerto Rico-Mayagüez, Box 9000, Mayagüez, PR 00681, USA; elvin.roman@upr.edu (E.R.-P.); alejandro.segarra@upr.edu (A.E.S.-C.); consuelo.estevez@upr.edu (C.E.d.J.); dania.rivera@upr.edu (D.R.-O.)
\* Correspondence: rebecca.tirado@upr.edu or rebeccatiradocorbala@gmail.com; Tel.: +1-(787)-370-9179

**Abstract:** In Puerto Rico, Huanglongbing (HLB), detected in 2009, continues to produce losses in orchards across the island. Efforts to produce clean propagation materials and select different combinations of scions and rootstocks to mitigate the disease have been a priority. In 2016, an experiment was established in Isabela in Coto clay series soils at 120 m to assess 'Mexican' lime, 'Fina' clementine mandarin, and 'Campbell' Valencia orange grafted in 'Swingle' citrumelo, 'HRS 812', and 'HRS 897'. Tree growth parameters, tree fruit production, HLB incidences, and tree mortality were evaluated under a fertigation system. Higher fruit production was obtained with 'Fina' clementine mandarin and 'Campbell' Valencia orange when grafted in 'HRS 812' followed by 'HRS 897'. 'Mexican' lime had higher fruit production when grafted in 'HRS 897'. On the basis of total fruit production and HLB incidence, 'HRS 812' was outstanding, yielding fruits with higher values even though they were infected with HLB. Moreover, 'HRS 897' rootstock was demonstrated to be a potential rootstock for the Puerto Rico conditions.

**Keywords:** Oxisol; 'Mexican' lime; 'Fina' clementine mandarin; 'Campbell' Valencia orange; rootstocks; scions; fertigation

## 1. Introduction

In Puerto Rico, as in other parts of the world, citrus is among the most economically important fruit crops. In 2017, citrus production had a net value of 8.6 million USD, as reported by the Department of Agriculture of Puerto Rico (DAPR) [1,2]. On the island, oranges and mandarins are consumed primarily as fresh fruit or as juice and are mainly grown in middle- to high-elevation locations (100–800 m above sea level). Other citrus fruits, such as lemons and limes, are grown mainly near the coast [3].

Unfortunately, most citrus production in Puerto Rico is currently vulnerable to Huanglongbing (HLB; also known as citrus greening) disease (*Candidatus* Liberibacter asiaticus (*CLas*)). The disease is vectored by the Asian citrus psyllid (ACP), *Diaphorina citri* Kuwayama, an insect introduced in 2001 and found commonly at low to middle elevations (0–600 m) [3–6]. Differential susceptibility has also been observed between citrus types and varieties, and losses from HLB appear to be more severe in orange and mandarin than in lime or lemon orchards [5].

Successful citrus production in Puerto Rico may now depend on conducting integrative research that emphasizes (a) the use of different scion–rootstock combinations, adaptable to a wide range of environmental conditions [3,4], and (b) the adoption of optimal crop management practices, especially regarding irrigation and fertilization regimes [2]. This is due to the fact that, under HLB pressure, citrus requires high fertilization and irrigation levels, primarily during the early growth stages [2,7–9]. Similarly, research in identifying

high-quality rootstocks is needed. To date, only a handful of rootstocks are commonly used by farmers [3,4,9–11], which were mainly scrutinized before the advent of HLB in Puerto Rico.

'Swingle' citrumelo (*C. paradise* Macf. × *P. trifoliata* (L.) Raf.) and 'HRS 812' ('Sunki' mandarin, *C. reticulata* × *P. trifoliata* (L.) Raf.) are among those rootstocks tested and recommended to Puerto Rico farmers after the HLB appearance. 'HRS-897' ('Cleopatra' mandarin, *C. reticulata blanco* × *P. trifoliata* (L.) Raf.) is a new rootstock under evaluation on the island. According to Castle et al. [12], the responses of the previously mentioned rootstocks to soil conditions such as high clay content and diseases such as high HLB incidence are the two constraints primarily found in Isabela, Puerto Rico. 'Swingle' adapts poorly in soils with high clay content and has an intermediate response to HLB incidence, while 'HRS 812' has an intermediate response in soils with high clay content and intermediate HLB incidence. However, 'HRS 897' has a good adaptation to highly clay soils and intermediate HLB incidence. 'HRS 897' tends to have small trees, while 'HRS 812' and 'Swingle' have trees of intermediate size [12]. However, some trees grown in 'HRS 812' rootstocks produce good yields under high HLB pressure and exhibit lower-than-average rates of fruit drop.

Accordingly, the objective of this publication is to present the results of the early response of integrated field assays, conducted under natural HLB pressure, which describe the effects of optimal fertigation on soil fertility, tree growth, leaf nutrition, and yield in new orchard plantings of three citrus varieties ('Mexican' lime, 'Fina' clementine mandarin, and 'Campbell' Valencia orange) each grafted in one of three selected rootstocks ('HRS 812', 'HRS 897', and 'Swingle').

## 2. Materials and Methods

### 2.1. The Experimental Area

The experimental orchards were sown in December 2016 at the Agricultural Experiment Substation (AES) at Isabela (Figure 1), which is located in the northwest of the island of Puerto Rico (18.46° N and 67.05° W) at 120 m above sea level. This citrus orchard was established on a Coto series (*very fine, kaolinitic, isohyperthermic, Typic Hapludox*) soil with 7.80% sand, 10.0% silt, and 82.2% clay [13]. The annual average precipitation is 1639 mm, with May as the rainiest month and February as the driest. The maximum average temperature is $29 \pm 6$ °C [4].

### 2.2. Scion–Rootstock Combinations, Disease Testing, and Orchard Management

'Mexican' lime (*Citrus aurantifolia*), 'Fina' clementine mandarin (*Citrus clementina* Hort. Ex Tanaka), and 'Campbell' Valencia orange (*Citrus sinensis*) scions were grafted in three different citrus rootstocks: (1) 'Swingle' citrumelo (*C. paradise* Macf. × *P. trifoliata* (L.) Raf.), (2) 'HRS 812' ('Sunki' mandarin, *C. reticulata* × *P. trifoliata* (L.) Raf.), and (3) 'HRS-897' ('Cleopatra' mandarin, *C. reticulata blanco* × *P. trifoliata* (L.) Raf.). 'Rhode Red' Valencia orange trees grafted in 'Swingle' rootstock were used as borders. Rootstock trees were produced from seeds; they were sown in November 2015 and, around 4 months (March–April 2016) after, were grafted. The citrus plants were grown in a screenhouse protected with anti-insect screen mesh (0.24 mm × 0.75 mm) up to ~0.61 m in height. At planting time (2016), and in 2018 and 2019, trees were tested for the presence of '*Candidatus Liberibacter asiaticus*' using isothermal amplification, following the manufacturer's instructions (Envirologix, Inc., Portland, ME, USA). This method uses the commercial kit 'leaf petiole DNable for Citrus Greening' (Cat. No. DF-02-PT, Envirologix Inc., Portland, ME, USA), and the amplification is displayed in an AmpliFire™. During the years 2020 and 2021, the detection of Ca. *Liberibacter asiaticus* was conducted by DNA amplification using conventional polymerase chain reaction (PCR) [14] with primers OI1 and OI2c (IDT Technologies, Coralville, IA, USA). Three petioles from each tree were collected for the conventional PCR assay, and positive and negative control samples were included.

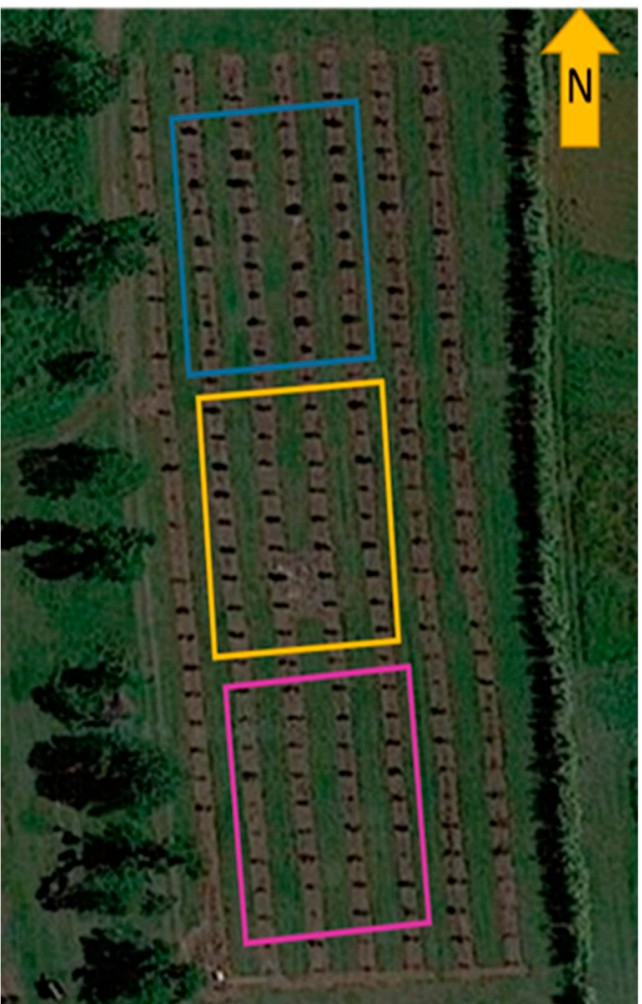

**Figure 1.** View of the experimental orchard established in the Agricultural Experiment Substation of Isabela. The blue rectangle represents 'Mexican' lime orchard, the yellow rectangle is planted with 'Fina' clementine mandarin, and the pink rectangle represents 'Campbell' Valencia orange. Border trees are 'Rhode red' Valencia orange grafted in 'Swingle' citrumelo.

Experimental Orchard Design

Tested scion–rootstock combinations in the experimental orchards were arranged in a randomized complete block design (RCBD) with four replicates. Trees were sown at a distance of 4.5 by 5.9 m, with each experimental plot containing three trees. A supplementary drip irrigation system was installed, and water was provided as needed on the basis of volumetric water content and precipitation recorded in the field. Each tree had two emitters calibrated for 4 gals/h each. The three citrus species received the same management practices. The trees were fertigated on the basis of their nutritional requirements (<5 years old) [15,16]. Citrus trees growing in soil with good drainage with a pH of 6.0–7.5 require 0.08 kg/tree/year of each element (N–P–K).

From January 2017 to October 2021, the following fertilizers were supplied through the irrigation system: Nitro 30 (30–0–0) (each tree received 3.49 fl. oz. six times a year); STARTER PLUS (8–32–5) (each tree received 1.0 fl. oz. twice a year); RECOVER (3–18–18) (each tree received 0.5 fl. oz. twice a year); Probalance (15–2–15) (each tree received 0.4 fl. oz. once a year). Furthermore, control of insect vectors and other pests was achieved by applying the systemic insecticide Imidacloprid (Alias 4F®) through the irrigation system twice a year (from March to November when high psyllid populations are common) at a total rate of 16 fl. oz. acre and through foliar applications of Abamectin (Abba®) and Thyme Guard four times a year at a rate of 3 fl. oz. per acre each. Glyphosate (Roundup®)

was applied for weed control as needed at 0.75–1.5 lbs. AI per acre, depending on the stage of maturity of weeds.

### 2.3. Soil Fertility, Tree Growth, Leaf Nutrient Analysis, and Yield

To determine soil fertility, composite soil samples were collected from the upper 20 cm around each citrus cultivar/rootstock combination, using a 7.62 cm bucket auger on a yearly basis. Soil pH was measured in a 1:1 (*v:v*) soil–water mixture [17]. Exchangeable calcium ($Ca^{2+}$), magnesium ($Mg^{2+}$), sodium ($Na^+$), and potassium ($K^+$) were extracted using 1 M $NH_4OAc$ [18] and available phosphorus (P) by Olsen extracts. Organic matter (OM) was determined by loss on ignition. Total S was determined via inductively coupled plasma spectrometry (ICP) (Teledyne Leeman Labs Prodigy Dual, Hudson, NH, USA) after perchloric acid digestion [19]. Nitrate ($NO_3$-N) content (1:1 soil:distilled (DI) water) was determined using a Nitrate–Nitrite Astoria Pacific 2 analyzer (Portland, OR, USA). In addition, citrus leaf tissue samples were taken from the central part of trees at four coordinates—north, south, east, and west—were collected simultaneously from each scion–rootstock combination, replicated, and analyzed for N, Ca, Mg, P, K, manganese (Mn), iron (Fe), copper (Cu), boron (B), aluminum (Al), Na, and zinc (Zn), extracted using Mehlich 3.

Tree growth variables were measured in 2019–2021, including height, diameter, and canopy volume, to determine crop performance. Tree height and diameter were measured using a telescoping measuring pole, and total canopy volume (CV) was calculated using the Fallahi and Mousavi [20] equation: $CV = 0.524 \times$ tree height (m) $\times$ tree square diameter ($m^2$). The yield efficiency was calculated using the average fruit number divided by CV. Fruit yield variables, fruit number, and size were totaled for 2018–2021. In 2021, fruit yield variables were measured up to October 2021. Fruit production (fruit number and size) was quantified at least eight times each year (from February to December).

### HLB Detection Using Conventional PCR

In each sample, total DNA was isolated from 200 mg of finely ground leaf midribs. The tissue was placed in a 1.5 mL Eppendorf tube with four glass beads and pulverized using a mini bead beater Biospec 3110BX. The DNA was isolated using a DNeasy Plant mini kit (Qiagen, Valencia, CA, USA) following the manufacturer's instructions with modifications recommended by USDA, APHIS, PPQ, and CPHST [21]. Conventional polymerase chain reaction (PCR) was performed with each DNA sample. The PCR mix was composed of 12.5 μL of Go Taq® Master Mix (Promega, Madison, WI, USA, Cat No. M712B), 2.5 μL of water, and 200 nM of primers OI1/OI2c proposed by Li et al. [14] for the amplification of the 16S rDNA bacterial DNA region. To each PCR reaction, 25 ng of total plant DNA was added for a total volume of 25 μL. The amplification program started with 2 min at 94 °C, followed by 35 cycles of 30 s at 94 °C, 30 s at 62 °C, and 1 min at 72 °C. The PCR was performed using a thermocycler Biometra T3000 (Labrepco, Göttingen, Germany).

### 2.4. Statistical Analysis

Analysis of variance (ANOVA), followed by means separation using Tukey's honestly significant difference test at $\alpha < 0.05$ for an RCBD design, was used to compare soil fertility, tissue analyses, tree and yield variables, and HLB prevalence from different scion–rootstock combinations. Statistical analysis was undertaken using JMP Version 10 (SAS Institute, Cary, NC, USA).

## 3. Results

### 3.1. Soil Chemical Properties

Soil OM, K, Mg, Na, P, S, and $NO_3$-N concentrations varied significantly by rootstock. In 'Mexican' lime scions, soil collected where 'HRS 897' rootstocks grew had higher OM, K, Mg, Na, and P concentrations than 'HRS 812' and 'Swingle' (Table 1). For 'Fina' clementine mandarin scions, lower P concentrations were found in soil collected from 'HRS 812' compared with 'HRS 897' and 'Swingle' rootstocks. In 'Campbell' Valencia orange, lower

concentrations of K, Na, P, S, and $NO_3$-N were found in soil collected from 'Swingle' rootstock than 'HRS 812' and 'HRS 897'. No significant differences in the remaining studied variables were detected for each citrus variety (Table 1).

**Table 1.** Soil nutrients in 2021 of three citrus cultivars grafted in three different rootstocks and growing under fertigation practices on Coto series soil at the Agricultural Experiment Station of Isabela, Puerto Rico.

| Scion | Rootstocks | OM [z] | pH | Ca | K | Mg | Na | P | S | $NO_3$-N |
|---|---|---|---|---|---|---|---|---|---|---|
| | | % | 1:1 | | | | $mg \cdot kg^{-1}$ | | | | ppm |
| 'Mexican' lime | 'Swingle' | 3.68 [b,y] | 6.55 [a] | 930 [a] | 95.5 [b] | 91.1 [b] | 12.0 [b] | 11.0 [b] | 10.5 [a] | 2.5 [a] |
| | 'HRS 812' | 3.75 [b] | 6.38 [a] | 909 [a] | 95.0 [b] | 90.3 [b] | 12.2 [b] | 11.5 [b] | 9.75 [a] | 2.5 [a] |
| | 'HRS 897' | 4.38 [a] | 6.38 [a] | 846 [a] | 129 [a] | 103 [a] | 14.4 [a] | 14.8 [a] | 10.8 [a] | 3.0 [a] |
| 'Fina' clementine mandarin | 'Swingle' | 4.92 [a] | 7.38 [a] | 1595 [a] | 88.6 [a] | 73.7 [a] | 12.2 [a] | 13.8 [a] | 7.75 [a] | 2.5 [a] |
| | 'HRS 812' | 4.63 [a] | 6.98 [a] | 1422 [a] | 84.0 [a] | 76.9 [a] | 13.0 [a] | 11.3 [b] | 8.25 [a] | 2.5 [a] |
| | 'HRS 897' | 4.78 [a] | 7.38 [a] | 1832 [a] | 75.2 [a] | 67.1 [a] | 11.9 [a] | 14.0 [a] | 7.25 [a] | 2.75 [a] |
| 'Campbell' Valencia orange | 'Swingle' | 5.43 [a] | 7.73 [a] | 1838 [a] | 69.6 [b] | 55.5 [a] | 9.93 [b] | 8.75 [b] | 6.75 [b] | 2.25 [b] |
| | 'HRS 812' | 5.28 [a] | 7.58 [a] | 1930 [a] | 115 [a] | 54.8 [a] | 12.3 [a] | 11.8 [a] | 7.85 [a] | 3.0 [a] |
| | 'HRS 897' | 5.44 [a] | 7.40 [a] | 1698 [a] | 96.3 [a] | 65.7 [a] | 12.4 [a] | 12.0 [a] | 9.0 [a] | 3.5 [a] |

[z] OM = organic matter, Ca = exchangeable calcium, K = exchangeable potassium, Mg = exchangeable magnesium, Na = exchangeable sodium, P = available phosphorus, S = total sulfur, and $NO_3$-N = nitrate. [y] Means followed by the same letter in a column within each scion are not significantly different according to Tukey's test at $\alpha < 0.05$.

### 3.2. Leaf Tissue Macronutrient and Trace Element Concentrations

Leaf tissue N, K, and P concentrations varied significantly by rootstock/scion combinations in the three citrus varieties (Table 2). Ca, Na, and S concentrations only ranged significantly by rootstock in 'Mexican' lime and 'Fina' clementine mandarin cultivars. For N, lower concentrations were found in 'Mexican' lime and 'Campbell' Valencia orange grafted in 'Swingle' rootstock than grafted in 'HRS 812' and 'HRS 897'. No statistical difference was found between rootstocks for 'Fina' clementine mandarin (Table 2). For Ca, lower concentrations were found in 'Mexican' lime and 'Fina' clementine mandarin grafted in 'HRS 897' rootstock than 'Swingle' and 'HRS 812' (Table 2). For K, lower concentrations were found in 'Mexican' lime and 'Fina' clementine mandarin grafted in 'Swingle' rootstock than grafted in 'HRS 812' and 'HRS 897'. However, lower concentrations for 'Campbell' Valencia orange were found when grafted in 'HRS 812' than 'HRS 897' and 'Swingle'. For 'Campbell' Valencia orange, no statistical differences were found for Ca, Na, and S elements. No statistically significant difference was found for Mg in all scion–rootstock combinations (Table 2). For Na, lower concentrations were found in 'Mexican' lime and 'Fina' clementine mandarin grafted in 'Swingle' rootstock than grafted in 'HRS 812' and 'HRS 897'. For S, lower concentrations were found in 'Mexican' lime and 'Fina' clementine mandarin grafted in 'HRS 897' than grafted in 'Swingle' and 'HRS 812' rootstocks. For the three cultivars, higher P concentrations were found in 'Swingle' rootstock than grafted in 'HRS 812' and 'HRS 897'.

Leaf tissue Al and B concentrations varied significantly by rootstock in the three citrus varieties (Table 3). Fe concentrations ranged significantly by rootstock only in 'Mexican' lime. Tissue collected from 'HRS 812' in the three scions had higher B concentrations than those collected from 'Swingle' and 'HRS 897'. No significant difference was recorded in tissue concentrations of Cu, Mn, and Zn.

### 3.3. Tree Growth and Yield

Yearly measurements of tree height, root-to-shoot ratio, canopy volume, and tree efficiency, observed between 2019 and 2021, are shown in Table 4. For 'Mexican' limes, in the year 2019, significant differences were observed in canopy volume and tree efficiency variables. Higher canopy volumes were found in 'HRS 897' rootstock compared with the other combinations. However, higher tree efficiency was found when grafted in 'HRS 812'

followed by 'HRS 897' and 'Swingle'. In the year 2020, higher canopy volume was also found in 'HRS 897' compared with the other two rootstocks. Furthermore, higher tree efficiency was found for 'HRS 897' followed by 'HRS 812' and 'Swingle'. In the year 2021, higher tree height, canopy volume, and tree efficiency were found in 'Swingle' than the other two rootstocks. Higher total yield (number, #) was found in 'HRS 897' followed by 'HRS 812', with the least production in 'Swingle' (Table 4).

**Table 2.** Mean concentrations of leaf tissue nutrients in 2021 of three citrus cultivars grafted in three different rootstocks and growing under fertigation practices on Coto series soil at the Agricultural Experiment Station of Isabela, Puerto Rico.

| Scion | Rootstocks | N [z] | Ca | K | Mg | Na | P | S |
|---|---|---|---|---|---|---|---|---|
| | | % | | | mg·kg$^{-1}$ | | | |
| 'Mexican' lime | 'Swingle' | 2.09 [b] | 3.52 [a] | 1.15 [b] | 0.263 [a] | 0.108 [b] | 0.245 [a] | 0.290 [a] |
| | 'HRS 812' | 2.31 [a] | 3.58 [a] | 1.62 [a] | 0.243 [a] | 0.135 [a] | 0.183 [b] | 0.250 [a] |
| | 'HRS 897' | 2.30 [a] | 2.92 [b] | 1.50 [a] | 0.225 [a] | 0.145 [a] | 0.173 [b] | 0.200 [b] |
| 'Fina' clementine mandarin | 'Swingle' | 1.97 [a] | 4.05 [a] | 1.19 [b] | 0.263 [a] | 0.045 [b] | 0.318 [a] | 0.250 [a] |
| | 'HRS 812' | 2.09 [a] | 4.12 [a] | 1.45 [a] | 0.263 [a] | 0.078 [a] | 0.290 [b] | 0.238 [a] |
| | 'HRS 897' | 2.16 [a] | 3.37 [b] | 1.35 [a] | 0.285 [a] | 0.085 [a] | 0.292 [b] | 0.200 [b] |
| 'Campbell' Valencia orange | 'Swingle' | 2.11 [b] | 3.66 [a] | 1.51 [a] | 0.233 [a] | 0.168 [a] | 0.315 [a] | 0.175 [a] |
| | 'HRS 812' | 2.30 [a] | 3.40 [a] | 1.27 [b] | 0.223 [a] | 0.183 [a] | 0.270 [b] | 0.189 [a] |
| | 'HRS 897' | 2.33 [a] | 3.43 [a] | 1.58 [a] | 0.232 [a] | 0.163 [a] | 0.280 [b] | 0.200 [a] |

[z] N = nitrogen, Ca = calcium, K = potassium, Mg = magnesium, Na = sodium, P = phosphorus, and S = sulfur. Means followed by the same letter in a column within each scion are not significantly different according to Tukey's test at $\alpha < 0.05$.

**Table 3.** Mean concentrations of plant micronutrients and trace elements in 2021 of three citrus cultivars grafted in three different rootstocks and growing under fertigation practices on Coto series soil at the Agricultural Experiment Station of Isabela, Puerto Rico.

| Scion | Rootstocks | Al [z] | B | Cu | Fe | Mn | Zn |
|---|---|---|---|---|---|---|---|
| | | % | | | mg·kg$^{-1}$ | | |
| 'Mexican' lime | 'Swingle' | 37.3 [b] | 58.8 [b] | 4.50 [a] | 121 [b] | 59.3 [a] | 11.0 [a] |
| | 'HRS 812' | 44.0 [a] | 86.3 [a] | 4.75 [a] | 190 [a] | 48.5 [a] | 18.8 [a] |
| | 'HRS 897' | 36.0 [b] | 79.3 [a] | 4.25 [a] | 140 [b] | 50.0 [a] | 14.3 [a] |
| 'Fina' clementine mandarin | 'Swingle' | 36.5 [b] | 94.3 [a] | 7.00 [a] | 141 [a] | 53.5 [a] | 13.8 [a] |
| | 'HRS 812' | 49.0 [a] | 70.3 [b] | 5.50 [a] | 119 [a] | 53.0 [a] | 12.5 [a] |
| | 'HRS 897' | 37.8 [b] | 89.0 [a] | 5.50 [a] | 104 [a] | 42.0 [a] | 12.3 [a] |
| 'Campbell' Valencia orange | 'Swingle' | 37.8 [b] | 98.8 [a] | 4.50 [a] | 77.3 [a] | 36.0 [a] | 9.50 [a] |
| | 'HRS 812' | 43.8 [a] | 71.8 [b] | 3.50 [a] | 66.0 [a] | 40.5 [a] | 8.75 [a] |
| | 'HRS 897' | 37.8 [b] | 94.8 [a] | 4.00 [a] | 80.0 [a] | 38.0 [a] | 8.50 [a] |

[z] Al = aluminum, Bo = Boron, Cu = Cooper, Fe = Iron, Mn = manganese, and Zn = zinc. Means followed by the same letter in a column within each scion are not significantly different according to Tukey's test at $\alpha < 0.05$.

For 'Fina' clementine mandarin in the years 2019–2020, significant differences were observed in canopy volume and tree efficiency. Higher canopy volume was found in 'HRS 897' compared with the other two rootstocks in both years. However, higher tree efficiency was found when grafted in 'HRS 897' followed by 'HRS 812' and 'Swingle' in the year 2019. In 2020, higher tree efficiency was observed in 'HRS 812' and 'HRS 897' than in 'Swingle'. In the year 2021, higher tree height and lower tree efficiency were found for 'Swingle' than the other two rootstocks in the year 2020. Higher tree efficiency was obtained when 'Fina' clementine mandarin was grafted in 'HRS 812'. Higher total yield (#) was found in 'HRS 812' followed by 'HRS 897', with the least production in 'Swingle' (Table 4).

**Table 4.** Tree height, root-to-shoot ratio, canopy volume, efficiency, and total yield (2018–2021) of three citrus cultivars grafted in three different rootstocks growing on Coto soil series at the Agricultural Experiment Substation of Isabela, Puerto Rico in 2019–2021.

| Scion | Rootstocks | Tree Height (m) | | | Root: Shoot Ratio | | | Canopy Volume (m³) | | | Tree Efficiency (Fruits m⁻³) | | | Total Fruit |
|---|---|---|---|---|---|---|---|---|---|---|---|---|---|---|
| | | *2019* | *2020* | *2021* | *2019* | *2020* | *2021* | *2019* | *2020* | *2021* | *2019* | *2020* | *2021* | (#) |
| 'Mexican' lime | 'Swingle' | 1.73 ᵃ | 1.80 ᵃ | 2.28 ᵃ,ʸ | 50.0 ᵃ | 75.6 ᵃ | 90.8 ᵃ | 6.40 ᵇ | 6.66 ᵇ | 8.43 ᵃ | 5.61 ᶜ | 5.21 ᶜ | 9.78 ᵃ | 143 ᶜ |
| | 'HRS 812' | 1.79 ᵃ | 1.91 ᵃ | 2.06 ᵇ | 54.2 ᵃ | 63.9 ᵃ | 110 ᵃ | 6.50 ᵇ | 6.94 ᵇ | 7.48 ᵇ | 6.71 ᵃ | 6.39 ᵇ | 8.93 ᵇ | 181 ᵇ |
| | 'HRS 897' | 1.72 ᵃ | 1.85 ᵃ | 1.93 ᶜ | 54.0 ᵃ | 84.7 ᵃ | 96.4 ᵃ | 6.90 ᵃ | 7.42 ᵃ | 7.74 ᵇ | 6.22 ᵇ | 9.16 ᵃ | 7.80 ᶜ | 212 ᵃ |
| 'Fina' clementine mandarin | 'Swingle' | 2.09 ᵃ | 2.12 ᵃ | 2.40 ᵃ | 47.6 ᵃ | 85.5 ᵃ | 91.4 ᵃ | 7.69 ᵇ | 7.80 ᵇ | 8.83 ᵃ | 6.99 ᶜ | 9.74 ᵇ | 12.0 ᶜ | 259 ᶜ |
| | 'HRS 812' | 2.00 ᵃ | 2.05 ᵃ | 2.18 ᵇ | 47.3 ᵃ | 82.2 ᵃ | 98.0 ᵃ | 7.63 ᵇ | 7.82 ᵇ | 8.32 ᵃ | 13.0 ᵇ | 21.7 ᵃ | 25.7 ᵃ | 580 ᵃ |
| | 'HRS 897' | 1.93 ᵃ | 1.98 ᵃ | 2.02 ᶜ | 47.1 ᵃ | 80.6 ᵃ | 85.5 ᵃ | 8.30 ᵃ | 8.52 ᵃ | 8.69 ᵃ | 14.3 ᵃ | 18.6 ᵃ | 20.2 ᵇ | 462 ᵇ |
| 'Campbell' Valencia orange | 'Swingle' | 1.28 ᶜ | 1.35 ᶜ | 1.89 ᵃ | 54.8 ᵃ | 77.9 ᵃ | 113 ᵃ | 4.12 ᶜ | 4.35 ᶜ | 6.08 ᵇ | 2.48 ᵃ | 7.98 ᵃ | 1.47 ᵇ | 44.5 ᵇ |
| | 'HRS 812' | 1.72 ᵇ | 1.74 ᵇ | 1.80 ᵃ | 51.5 ᵇ | 98.4 ᵃ | 135 ᵃ | 6.12 ᵇ | 6.19 ᵇ | 6.40 ᵇ | 1.06 ᵇ | 7.16 ᵃ | 6.51 ᵃ | 95.6 ᵃ |
| | 'HRS 897' | 1.84 ᵃ | 1.90 ᵃ | 1.95 ᵃ | 54.6 ᵃ | 86.2 ᵃ | 87.3 ᵇ | 7.67 ᵃ | 7.92 ᵃ | 8.13 ᵃ | 0.75 ᵇ | 8.59 ᵃ | 1.82 ᵇ | 34.9 ᵇ |

ʸ Means followed by the same letter in a column within each scion are not significantly different according to Tukey's test at α < 0.05.

For 'Campbell' Valencia orange, in the years 2019 and 2020, higher values of tree height and canopy volume were observed for trees grafted in 'HRS 897' followed by 'HRS 812' and 'Swingle'. However, in the year 2021, higher tree efficiency was found when 'Campbell' Valencia orange was grafted in 'HRS 812' rootstock. Higher total yield (#) was found in 'HRS 812', followed by 'HRS 897' and Swingle' with equal production (Table 4).

Figure 2 shows the total average fruit number, and Figure 3 shows the total fruit weight per tree of the three citrus varieties grafted in three different rootstocks between 2018 and 2021. Both studied variables had the same trend. For 'Mexican' lime, in the years 2018 and 2020, the higher total average number of fruits and total fruit weight per tree were found when grafted in 'HRS 897' compared to 'HRS 812' and 'Swingle'. For the years 2019 and 2021, no statistical difference was found between rootstocks. For 'Fina' clementine mandarin, in the year 2018, a higher average fruit number and total fruit weight per tree was found in 'HRS 812' rootstock. In the years 2019 and 2020, lower average fruit numbers and total fruit weight per tree were found when grafted in 'Swingle' than in 'HRS 812' and 'HRS 897'. In the year 2021, no statistical difference was found for the average fruit number and total fruit weight per tree between rootstocks. For 'Campbell' Valencia orange, in the years 2018–2019, higher fruit production (~2×) was found when grafted in 'Swingle'. In the years 2020–2021, higher average number of fruits and total fruit weight per tree were found when grafted in 'HRS 812' than in 'HRS 897' and 'Swingle' (Figures 2 and 3).

### 3.4. Number and Percentage of Trees Infected and Dead by HLB

The experimental field trees were tested on a yearly basis up to 2021. Once the trees tested positive for HLB, no further test was executed. Table 5 shows data on the total number and percentage of HLB infected trees and the number and percentage of dead trees. Tree death was also assumed to be caused by HLB infection. For 'Mexican' limes, a higher number of infected trees were observed in 'HRS 812' (75%) followed by 'Swingle' (41.7%) and 'HRS 897' (25%) (Table 5). However, no statistical differences were found between rootstocks for infected 'Fina' clementine mandarin and 'Campbell' Valencia orange trees. Regarding the number of dead trees, for 'Mexican' limes, dead trees were only observed when grafted in 'Swingle' rootstocks. Furthermore, dead trees were observed in 'Campbell' Valencia orange in 'Swingle' and 'HRS 897' rootstocks but not for 'Fina' clementine mandarin.

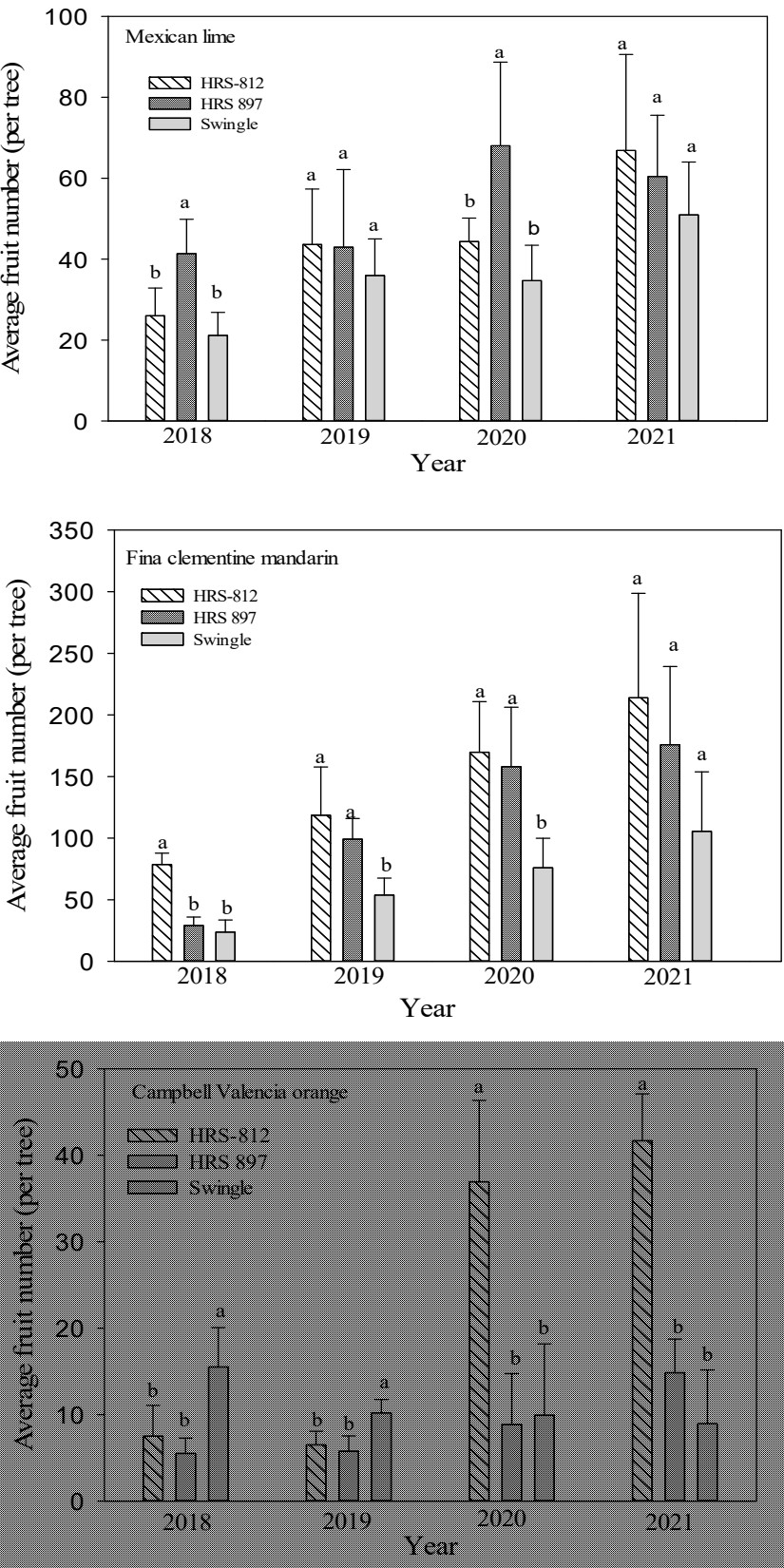

**Figure 2.** Total average fruit number (2018–2021) of 'Mexican' lime, 'Fina' clementine mandarin, and 'Campbell' Valencia orange grafted in three different rootstocks growing in Coto series soil at the Agricultural Experiment Substation of Isabela, Puerto Rico. Means ± standard error with different letters for each scion and year are not significantly different according to Tukey's test at $\alpha < 0.05$.

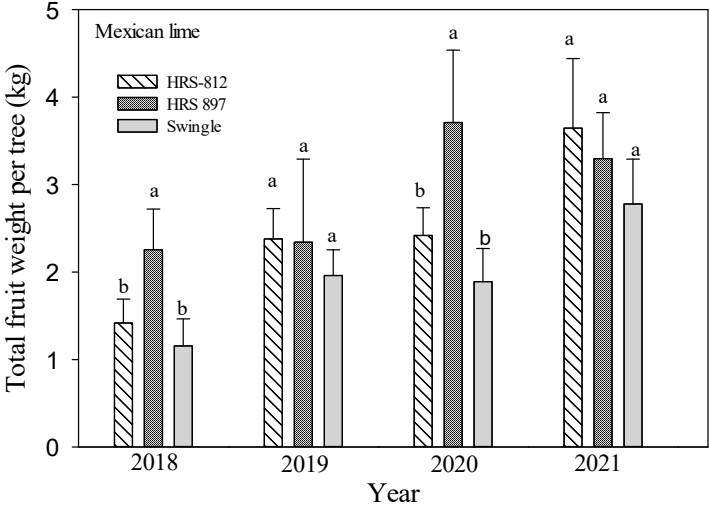

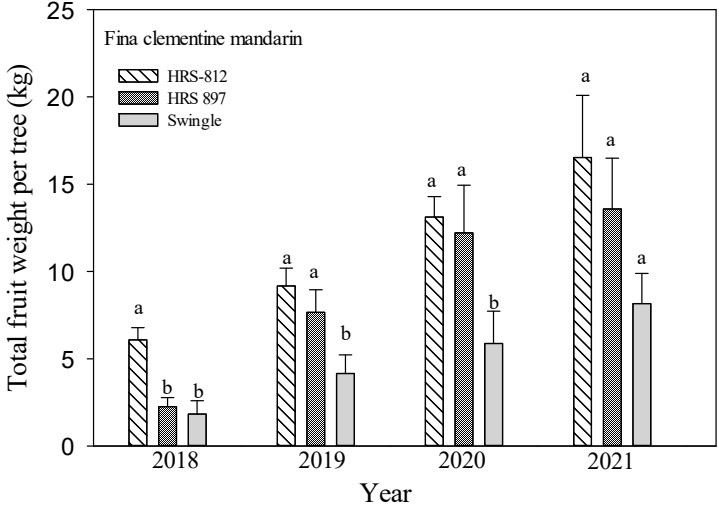

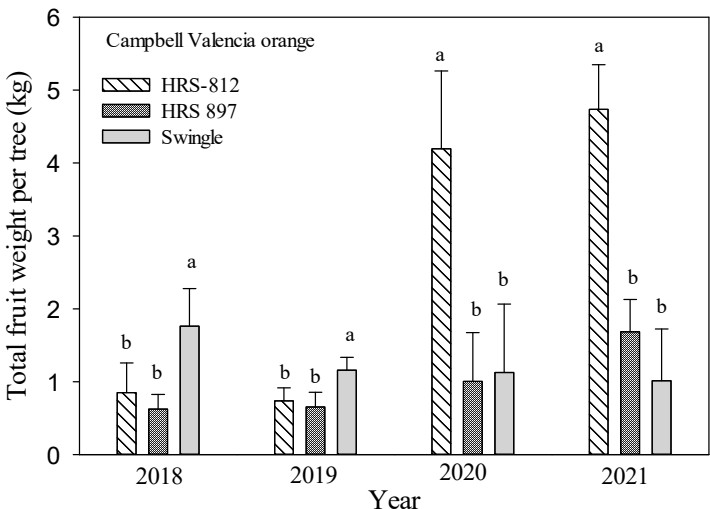

**Figure 3.** Total fruit weight in kg per tree (2018–2021) of 'Mexican' lime, 'Fina' clementine mandarin, and 'Campbell' Valencia orange grafted in three different rootstocks growing in Coto series soil at the Agricultural Experiment Substation of Isabela, Puerto Rico. Means ± standard error with different letters for each scion and year are not significantly different according to Tukey's test at $\alpha < 0.05$.

**Table 5.** Number and percentage of trees infected by and dead due to HLB in 2021.

| Scion | Rootstocks | Number of HLB Infected Trees | Percentage of HLB Infected Trees ^ | Number of Dead Trees | Percentage of Dead Trees ^ |
|---|---|---|---|---|---|
| | | (#) | (%) | (#) | (%) |
| **'Mexican' lime** | 'Swingle' | 5 [b,y] | 41.7 [b] | 2 [a] | 16.7 [a] |
| | 'HRS 812' | 9 [a] | 75.0 [a] | 0 [b] | 0 [b] |
| | 'HRS 897' | 3 [c] | 25.0 [c] | 0 [b] | 0 [b] |
| **'Fina' clementine mandarin** | 'Swingle' | 5 [a] | 41.7 [a] | 0 [a] | 0 [a] |
| | 'HRS 812' | 6 [a] | 50.0 [a] | 0 [a] | 0 [a] |
| | 'HRS 897' | 5 [a] | 41.7 [a] | 0 [a] | 0 [a] |
| **'Campbell' Valencia orange** | 'Swingle' | 7 [a] | 58.3 [a] | 3 [a] | 25.0 [a] |
| | 'HRS 812' | 8 [a] | 66.7 [a] | 0 [b] | 0 [b] |
| | 'HRS 897' | 5 [a] | 41.7 [a] | 2 [a] | 16.7 [a] |

[y] Means followed by the same letter in a column within each scion are not significantly different according to Tukey's test at $\alpha < 0.05$. ^ Percentages of HLB infected trees and dead trees were calculated on the basis of the total trees of each scion/rootstock (12) planted in the field.

## 4. Discussion

In our study, experimental orchards were established in Coto soils with a pH ranging from 6.38 to 7.73. Variability across the field (south to north; Figure 1) was due to tilled limestone carbonate outcrops during fields preparation before planting. In this case, our citrus trees were growing under ideal soil pH [4] with no expected yield reduction attributed to soil fertility.

Since HLB visual symptoms are similar to micronutrient deficiencies, accurate HLB detection is achieved through DNA tests [3,4]. Results from leaf analyses by Spann et al. [22] attributed nutrient deficiencies to HLB, where infected plants had significantly lower values of Zn, Fe, Mn, and Ca. In our study, according to Mills and Jones [16] and Zekri [23], all nutrients of the three studied scion/rootstock combinations (with the exception of Zn) were at adequate levels in plant tissue, indicating that deficiency visual symptoms were caused by HLB infection itself. Zn levels were deficient (lower than 25 mg/kg) in the three studied scion/rootstock combinations, especially in oranges [13,24,25]. However, according to Obreza and Morgan [24], our leaf N was deficient, while Mg may have been at low levels. To determine whether or not our leaf N, Mg, and Zn levels were deficient and affecting fruit production, each mentioned nutrient was correlated with 2021 total fruit production (data not shown). No positive correlation was observed for N ($R = 0.301$), Mg ($R = 0.099$), or Zn ($R = 0.254$) with respect to 2021 total fruit production.

On the basis of the total fruit production (2018–2021) versus HLB-infected trees for each scion–rootstock combination, these variables were directly correlated. For 'Mexican' lime, higher production was found when grafted in 'HRS 897'. For 'Fina' clementine mandarin and 'Campbell' Valencia orange, higher production was observed in 'HRS 812'. 'HRS 897' was demonstrated to be a new potential rootstock to be used by citrus growers, at least for lime. However, the 'HRS 812' rootstock was demonstrated to be the better choice since it has good production even though the trees were infected with HLB. Bowman and Rouse [25] found that the 'HRS 812' rootstock was highly productive in Florida with high-quality fruits and exhibited tolerance to Citrus tristeza virus (CTV) in laboratory trials. Wutscher and Bowman [26] found, using a Valencia sweet orange on 21 rootstocks, that 'HRS 812' was the first and second most productive rootstock through the fourth harvest season depending on the unit of yield measurement compared. This comparison tends to be clearer through the sixth harvest season when compared with other commercial rootstocks such as 'Swingle' and 'Carrizo'. Albrecht and Brown [27] found that 'HRS 897' and 'Carrizo' rootstocks were tolerant to HLB, while 'HRS 812' was considered moderately tolerant to HLB compared to Cleopatra mandarin. Furthermore, Albrecht and Brown [27,28] found that field-grown 'HRS 897' exhibited little tree damage despite prolonged pressure from

HLB. Bowman and Rouse [25] also described the 'HRS 812' rootstock to be tolerant to citrus blight and smaller in tree growth than other rootstocks, similar to our findings.

## 5. Conclusions

Both 'Fina' clementine mandarin and 'Campbell' Valencia orange growing on Coto clay series soil at 120 m showed higher fruit production when grafted in 'HRS 812', followed by 'HRS 897' and 'Swingle'. However, 'Mexican' lime had higher production when grafted in 'HRS 897'. According to our foliar analysis, all the nutrients of the three scion/rootstock combinations (with the exception of Zn) were likely under an adequate level. In the next year, the fertilization program will be adjusted for older trees, and the trees will receive a higher concentration of macronutrients plus the foliar application of micronutrients to cover Zn deficiencies. On the basis of production and tree infection with HLB, we recommend using the 'HRS 812' rootstock, which was demonstrated to be the better choice, followed by 'HRS 897', which was demonstrated to be a new potential rootstock to be used by citrus growers in Puerto Rico.

**Author Contributions:** R.T.-C. conceptualized the study and methodology and performed the statistical analysis of the variables; R.T.-C. wrote the original draft manuscript; E.R.-P. and A.E.S.-C. add some sections to the manuscript; C.E.d.J. performed the CLas, CTV, and CVC analysis of the trees before planting and during the experiment; C.E.d.J. provided data of HLB analysis; R.T.-C., E.R.-P., A.E.S.-C., C.E.d.J. and D.R.-O. review and edit the manuscript. All authors have read and agreed to the published version of the manuscript.

**Funding:** This research was supported by the United States Department of Agriculture, National Institute of food and Agriculture (NIFA), Hatch Program Project-94Q (Citrus plant production systems, genetic resources, and breeding).

**Institutional Review Board Statement:** Not applicable.

**Informed Consent Statement:** Not applicable.

**Data Availability Statement:** The data presented in this study are available on request from the corresponding author.

**Acknowledgments:** The authors would like to thank HATCH-94Q (citrus plant production systems, genetic resources, and breeding) for funding this study. In addition, the authors would like to thank Bryan Calero Dimaio and other personnel from the AES of Isabela and graduate students Gianna A. Fernández Soto, Carlos D. Barreda-Castro, and Lyvette Trabal for helping us with soil and plant sampling and field management.

**Conflicts of Interest:** The authors declare no conflict of interest.

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
