# Peer review of "Early Response of ‘Mexican’ Lime, ‘Fina’ Clementine Mandarin, and ‘Campbell’ Valencia Orange on Selected Rootstocks Grown under Fertigation Practices in an Oxisol in Puerto Rico"

_horticulturae, doi:10.3390/horticulturae8060513_

Round 1
Reviewer 1 Report
More information should have been given about the characteristics of rootstocks in the material section. It is stated in the research that the rootstocks were grafting in 2016.
In rootstock performance measurements, data for 2019 and 2020 are presented. Are the yield values of the rootstocks in the 3rd and 4th years sufficient, especially in terms of tree yield? clarification is required on this matter. What was the effect of rootstocks on fruit quality? Do researchers have an opinion on the resistance of rootstocks to biotic and abiotic stress conditions?
Author Response
Who main concern, thanks a lot for your suggestions. All the edits are in red color in the text. The edits cover the points (comments/suggestions) of three reviewers.
- More information should have been given about the characteristics of rootstocks in the material section. It is stated in the research that the rootstocks were grafting in 2016. I add more relevant information in section 2.2.
- In rootstock performance measurements, data for 2019 and 2020 are presented. Are the yield values of the rootstocks in the 3rd and 4th years sufficient, especially in terms of tree yield? I am not sure what are you are trying to say. I revise the manuscript and for some variables I provided three years data (2019 to 2021 for tree height, canopy volume, and tree efficiency) and from 2018-21 for average fruit number and total fruit weight per tree. So, I think is sufficient for an early response manuscript.
- clarification is required on this matter. What was the effect of rootstocks on fruit quality? Do researchers have an opinion on the resistance of rootstocks to biotic and abiotic stress conditions? See section 4 and 5.

Reviewer 2 Report
An interesting manuscript with new information on the cultivation of lime, clementine mandarin and Valencia orange.
However, the introduction should have included more information about rootstocks and their role.
Please change the purpose of the paper. Presenting a four-year study cannot be the goal because a four-year study is supposed to give an answer to a properly stated goal.
Please include the standard error in the tables as in the graphs.
The discussion lacks information as to why the rootstocks improved fruit quality. What mechanisms worked or could have worked
Author Response
Who main concern. Thanks a lot for your comments and suggestions. I incorporate them with the comments of another two reviewers. Below are some replies to your suggestions and comments. Also, I attached the manuscript with all the edits in red.
Thanks, Rebecca
1) However, the introduction should have included more information about rootstocks and their role. Done
2) Please change the purpose of the paper. Presenting a four-year study cannot be the goal because a four-year study is supposed to give an answer to a properly stated goal. Done.
3) Please include the standard error in the tables as in the graphs. I differ with the reviewer of including the std error in the tables. I already provide analysis of variance followed by means separation using Tukey’s Honestly Significant Difference test at α < 0.05. Most of the tables have a lot of information. If I add the std errors the tables will look to busy. In tables you should use one or the other not both things.
- The discussion lacks information as to why the rootstocks improved fruit quality. What mechanisms worked or could have worked. See section 4.

Reviewer 3 Report
The submitted paper titled “Early response of ‘Mexican’ lime, ‘Fina’ clementine mandarin and ‘Campbell’ valencia orange on selected rootstocks grown under fertigation practices in an Oxisol in Puerto Rico” is about a study to evaluate different rootstock/scion combinations in the lime orchards partly. The researchers are seaching for the best rootstocks for some selected scions.
I have just some short questions to increase the level of this great paper.
I think the readers are interested in the selected rootstocks involved to the trial. Their genetic background is well-done in the paper, but there have some unique characteristics, which are necessary to mention. It is possible to add a short description about the rootstocks? These very short descriptions can help for the readers to understand better the difference between involved rootstocks.
When did you count and weight the fruits to have the average fruit number?
Author Response
Who main concern. Thanks a lot for your comments and suggestions. I incorporate them with the comments of another two reviewers. Below are some replies to your suggestions and comments. Also, I attached the manuscript with all the edits in red.
Thanks, Rebecca
- It is possible to add a short description about the rootstocks? These very short descriptions can help for the readers to understand better the difference between involved rootstocks. Done
- When did you count and weight the fruits to have the average fruit number? The information is provided in Section 2.3 and is the following one- Fruit yield variables, fruit number, and size were totaled for 2018-2021. In 2021, fruit yield variables were measured up to October 2021. Fruit production (fruit number and size) was quantified at least eight times each year (From February to December).

Round 2
Reviewer 2 Report
I thank the authors for the changes made to the manuscript.
The changes made are satisfactory and have improved the quality of the manuscript.